# Effect of Dynamic Biaxial Loading of Car Seats

**Petr Lepsik \*, Vitezslav Fliegel and Ales Lufinka**

Faculty of Mechanical Engineering, Technical University of Liberec, Studentska 2, 461 17 Liberec, Czech Republic
\* Correspondence: petr.lepsik@tul.cz

**Abstract:** The aim of this article is to determine the effect of the biaxial loading of a car seat, in vertical and horizontal directions, on seating comfort. We measured the transfer characteristic in the vertical direction with a free load. The method of laboratory testing of the car seat was chosen due to the simple repetition of the tests, ensuring the same test conditions, with high accuracy of their reproduction. To determine the effect, tests with uniaxial loading were compared with tests with biaxial loading. The resulting characteristic was the transfer characteristic in the vertical direction. The effect of horizontal loading on the transfer characteristic of a car seat was determined.

**Keywords:** car seat; testing; biaxial loading; measurement standards; transfer characteristics





## 1. Introduction

Testing of car seats in laboratory conditions is performed according to the relevant standards [1–3]. Each standard specifies a test method that corresponds to a particular car seat load regime. The comfort of sitting and the level of fatigue after a long drive in the car depend on the interaction properties of the car seat with the human body at the point of contact with the seat. Reproduction of seat testing in real operation in laboratory conditions requires strict adherence to prescribed standards, i.e., simultaneous multi-axis loading in the vertical and two horizontal directions. Therefore, the test equipment must be increasingly sophisticated, enabling the implementation of load signals in multiple axes. Tests can be performed both in real operation and in the laboratory. In our development work, we deal with the improvement of laboratory equipment for testing car seats. We are based on the generation I device, which allowed only vertical loading of the seat [4]. The article describes a new test device of generation II, enabling biaxial loading of the seat. We are also preparing a third-generation test device [5] that will enable three-axis testing. Triaxial loading is suitable for full laboratory simulation of real car driving. The possibility of comparing tests from real operation and their uniform evaluation also depends on the method of realization of test signals. Of course, the signals must be recorded correctly in the actual driving of the car along the specified route. Seating comfort is influenced by the design of the seat [6], the real stiffness of the seat and backrest [7], the hardness of the PU foam filling [8], its height and width [9], the area of the driver's contact surface with the seat [10], etc. To determine comfort car seats, the transfer characteristic in the vertical direction [11] is commonly used. A number of studies have already been carried out in the area of loading PU foams describing their specific properties [8–10,12–15] and describing properties under uniaxial loading [16,17] using the uniaxial test equipment [4]. In the study [18], horizontal loading was also taken into account and the resonant frequency was sought in the fore-and-aft cross-axis. However, the load in the horizontal direction was studied separately, which according to studies [15,19,20], does not provide the desired results, and it is necessary to deal with the folded multi-axial loading. The aim of this paper is to compare the influence of one- and two-axes dynamic loading of car seats on the transfer characteristic in the vertical direction.

## 2. Materials and Methods

### 2.1. Testing Device

The current test facility was created as an innovation of the existing facility, which allowed the performance of load tests of car seats in only one axis, i.e., vertical in the "z" axis. However, current normative legislation requires testing in two axes, i.e., in the vertical axis and in the anteroposterior axis at the same time. In order to meet the requirements of the standard, we have added a horizontal actuator to the existing equipment, which serves as an exciter in the "x" axis (Figure 1). The excitation range in the vertical axis is plus/minus 200 mm, and the anteroposterior axis is plus/minus 50 mm. These ranges richly cover the requirements of currently valid standards as well as the ranges of measured values of excitation signals in real driving. A great advantage of the said test device is the overall energy consumption for the test. Because the exciters are electric, their consumption is an order of magnitude lower than that of their hydrodynamic analogist. A sufficient amount of oil with the required pressure is required for the hydrodynamic hexapod to function properly, but this is created in the unit with the required power. This increases the price of individual tests many times over. There is an expert discussion about the economy and ecology of the laboratory tests performed. Optimization of energy intensity of laboratory tests determines the possibilities of their practical use. The signals required by the test methodology, both measured and generated, will be used as test signals. The spatial movement of the seat or the load is then composed of the realized excitation signals. From the preferred fluctuations and frequencies of the excitation signal, we can prioritize the "force" of excitation in individual directions, which is not possible at hexapod [21].

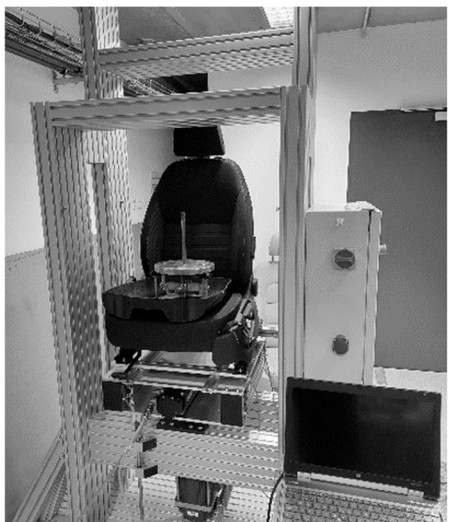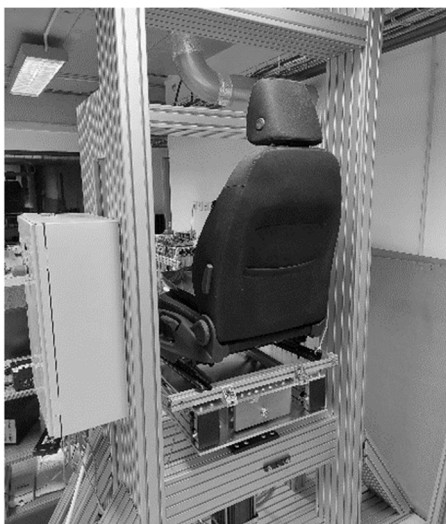

**Figure 1.** Original test equipment for biaxial loading.

### 2.2. Testing Loading

Fittings corresponding to the EuroSit III test dummy were used as test loads. The existing test equipment was used to perform the dynamic test. Loads differ in their design. The first type is very similar to the human body, i.e., it is shaped like a moderately static person. This manikin is used both in testing in a moving car as a "passenger" and in static testing of the seat, e.g., for measuring the H-point. The second type of load used is a European copy of a medium-sized statistical person (Figure 2). It is mainly used for dynamic tests, both with free load, e.g., for determining transfer characteristics, dynamic load, and with vertically guided load for determining, for example, seat creep. In this study, we used the second manikin.

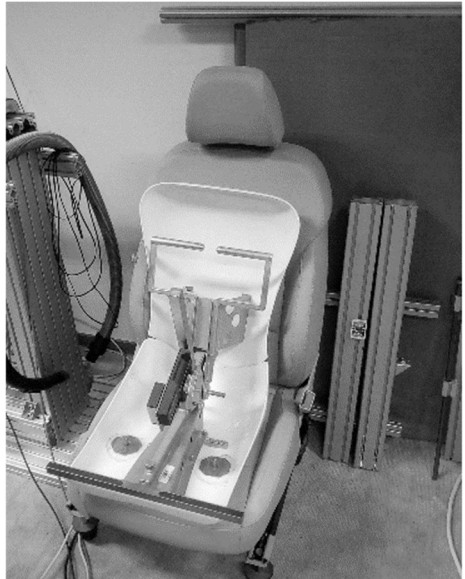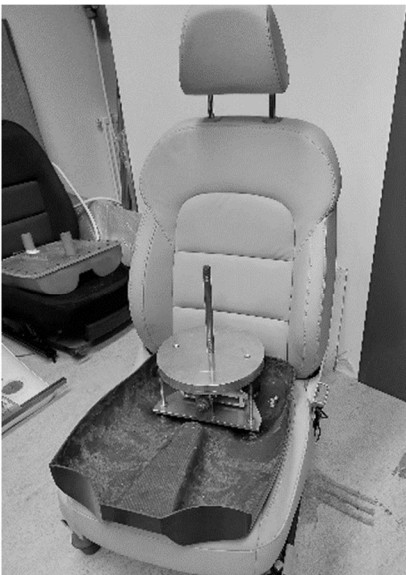

**Figure 2.** Original loading.

## 2.3. *Principles of Control*

The actuator is basically an electric motor with a gearbox and a motion screw. This converts the rotary motion of the motor into a sliding motion. The actuator is also equipped with a position sensor which is used for position feedback control. An integral part of the actuator is an external control unit that provides power to the electric motor and implements the position feedback control. The actuator control unit is connected to the user's computer in several ways. The RS 485 serial line and software supplied by the actuator manufacturer are used for basic parameter settings. A block diagram of the basic actuator connection is shown in Figure 3. The user application created in the Labview environment is then used to control the movement of the actuator during testing. It uses logical lines for communication, which are used for basic commands and status signals (e.g., start, ready, etc.). The value of the desired position of the actuator is transmitted by an analog signal, so the speed is not limited, for example, by the transfer speed of the serial line, and position changes can be very fast.

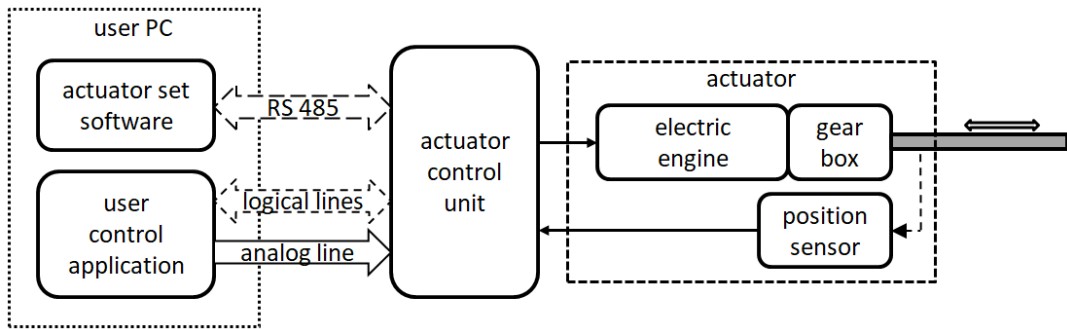

**Figure 3.** Block diagram of the basic actuator connection.

The test device allows movement in two axes Z and X. It, therefore, has two actuators, which are controlled simultaneously by the user application. A connection block diagram is shown in Figure 4.

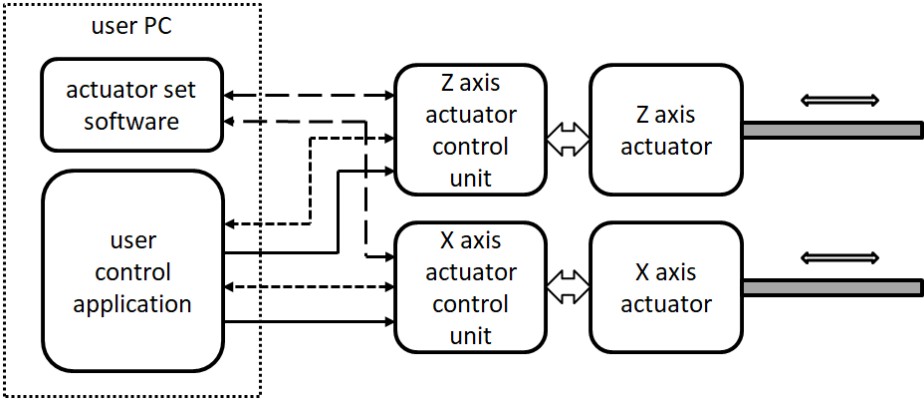

**Figure 4.** Block diagram of the testing device actuators connection.

The user control application can use an external file with a time record of the excitation signal for the actuator control or can directly generate fundamental harmonic signals. These methods can also be combined, for example, the Z axis can be controlled with an external file, and harmonic signals can be added to the X axis. This control method was also used for this testing, the Z axis was always excited by the same signal from the file and harmonic signals with different amplitudes and frequencies were gradually added to control the X axis.

*2.4. The Theoretical Basis of the Transfer Function Measurement*

The transfer function is generally defined as the ratio of the output and input functions of any device (Figure 5).

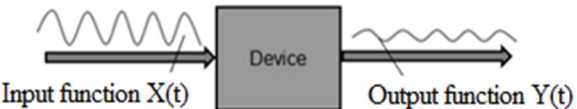

**Figure 5.** The principle of transfer function definition.

If the input function is defined as $X(t)$ and the output function as $Y(t)$, the transfer function $G(t)$ can be defined as their ratio.

$$G(t) = \frac{Y(t)}{X(t)} \tag{1}$$

Expressing the transfer function from the time course of the input and output functions is not very common, the image ratio obtained by the Fourier transformation of both functions is more often used.

$$G(j\omega) = \frac{Y(j\omega)}{X(j\omega)} \tag{2}$$

This representation of the transfer function is better for technical practice because it better describes the dependence of the transfer function on the frequency of the signals. The transfer function is a complex function, and its progress can, therefore, be displayed in the complex plane (Figure 6 left), which provides a comprehensive overview of the progress of the transfer depending on the frequency of the signal. For a clearer expression of the dependence of the transfer function on frequency, very often, instead of displaying it in a complex plane, the transfer function is divided into two parts—amplitude and phase, and each is displayed separately (Figure 6 right). The following relations are used to divide the transfer function.

$$G(j\omega) = A(j\omega)e^{j\varphi(\omega)} = P(\omega) + jQ(\omega) \tag{3}$$

$$A(\omega) = \sqrt{P^2(\omega) + Q^2(\omega)} \tag{4}$$

$$\varphi(\omega) = arctan \frac{Q(\omega)}{P(\omega)} \tag{5}$$

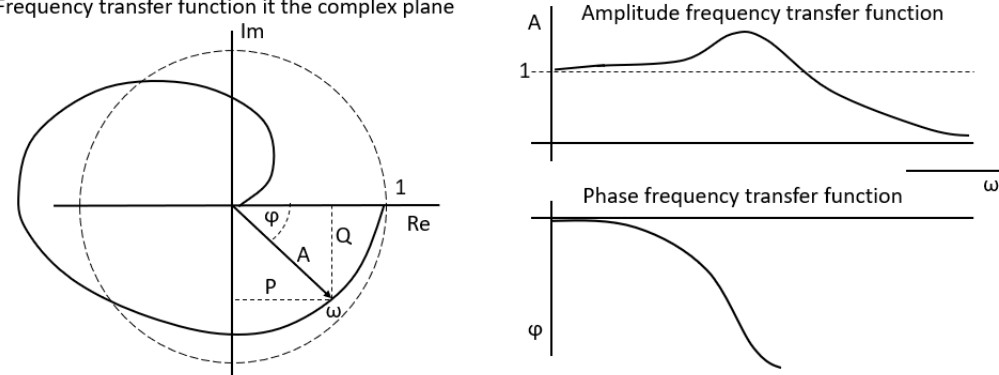

**Figure 6.** Different ways of displaying the transfer function.

Very often, only the amplitude-frequency characteristic is used to describe the properties of mechanical systems because the phase shift is not important from the point of view of the monitored criteria. This simplifies the description to one curve (Figure 6, top right). This is also how the basic description of the behavior of a car seat is expressed. The amplitude frequency characteristic expresses the damping properties of the car seat depending on the frequency of the excitation signal. If the curve amplitude transfer is less than 1, the car seat dampens the input oscillations. If it is greater than 1, the load (the person on the seat) goes into resonance. The maximum on the curve indicates the resonance frequency, i.e., the critical value of the frequency of the input oscillations, when the seating comfort is the worst because the vibration deflection is the largest.

*2.5. Test Signals and Measurement*

The resonant frequency of passive car seats is usually around 6 Hz. Therefore, an excitation signal in the range of 0.5 to 16 Hz is usually used for the measurement. The frequency increases continuously during the measurement in the specified range, and the amplitude of the oscillations is usually set so that the acceleration value of the excitation signal remains constant. Therefore, the amplitude must decrease with the increasing frequency of the excitation signal. Such a signal was created for the basic excitation in the Z axis, the value of acceleration amplitude was set to 0.1 G. Its example is shown in Figure 7.

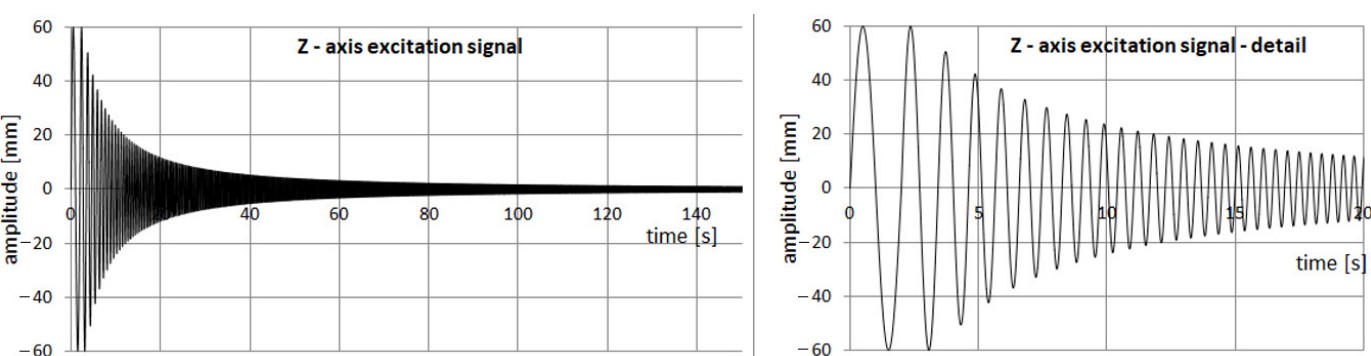

**Figure 7.** Test signal for Z-axis excitation.

Harmonic excitations in the X-axis direction with different amplitudes and frequencies were then added to this basic Z-axis excitation signal during testing. An example of such a compound excitation is shown in Figure 8.

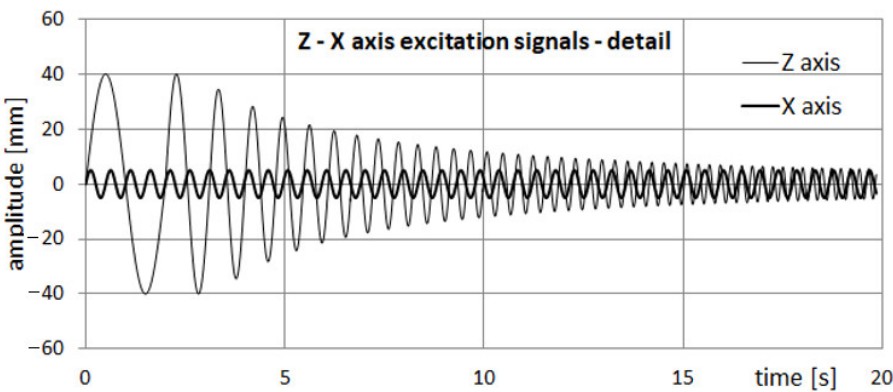

**Figure 8.** The example of the compound Z–X axis excitation.

The Dewe 5000 measuring device was used for the measurement. Two accelerometers measured the acceleration of the table and the mass in the direction of the Z axis. The experiment was further captured by a camera, the image recording was synchronized with the measured data. The block diagram of the measurement arrangement is in Figure 9. The sampling frequency of the data was 500 Hz, the camera took 100 images per second.

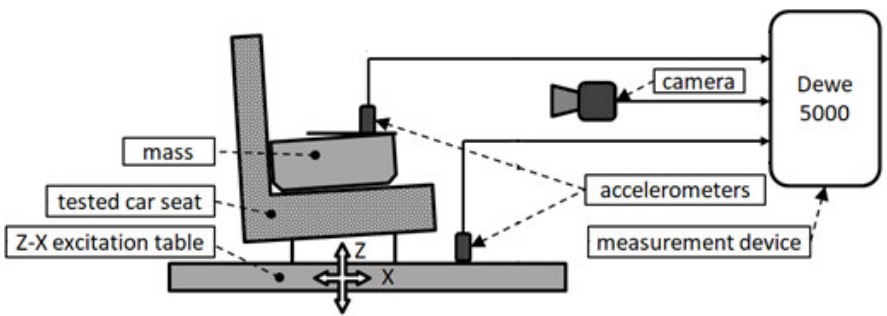

**Figure 9.** Block diagram of the measurement arrangement.

Mathematical channels for automatic calculation of the transfer function amplitude in real time were defined in the measuring device. In the first step, the amplitudes of the measured signals were detected, and the transfer value was calculated as the ratio of the amplitude of the signal from the mass and the signal of the excitation table. Two transfer functions were calculated, one from movements and the other from accelerations. The result is, therefore, two transfer functions, which should, however, be essentially the same. The double measurement and calculation method was chosen to refine the results and eliminate possible errors. In addition, the course of the transfer function amplitude is synchronized with the image recording. When analyzing the results, the shape of the characteristic can be assigned to the visible oscillations of the mass.

## 3. Results and Discussion

For comparison, tests were first performed with only vertical load in the Z axis in the frequency range from 1 to 16 Hz, which is the usual range for testing car seats. Figure 10 shows that resonance occurs in the region between 7 and 8 Hz. This completely corresponds with the characteristic features of the measured seat. For further experiments, the measured band was narrowed to a range from 3 to 10 Hz, which sufficiently covers the interesting region around the resonance frequency. By narrowing the band, the experiment time and data volume were significantly reduced. Increasing the low frequency further made it

possible to shorten the length of the window used for automatic amplitude detection. A narrower window allows for more accurate detection. Changing the detection parameters can cause even a small change in the calculated transfer function, so the results before and after the change cannot be compared. For all subsequent experiments, the settings remained the same so that the results were not affected and the effect of transverse excitation in the X-axis could be observed.

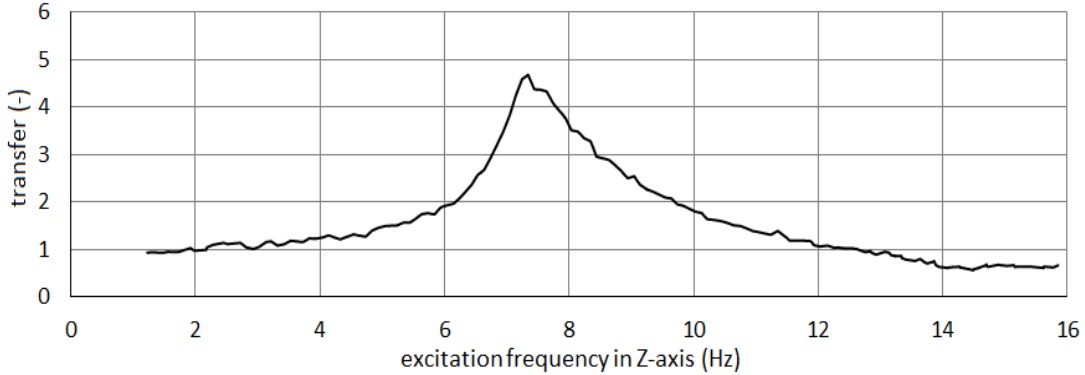

**Figure 10.** Transfer for Z-axis excitation 1 to 16 Hz without excitation in X-axis.

Furthermore, Figure 11 shows the transfer characteristics during simultaneous loading in both vertical and horizontal directions in selected control area of transfer changes from 3 to 10 Hz. The excitation frequency in the horizontal direction is 3 Hz. It can be seen from the transfer characteristics that both the resonance frequency and the resonance amplitude are reduced. In the interval from 3 Hz up to the resonance frequency, on the contrary, there is an increase in transfer.

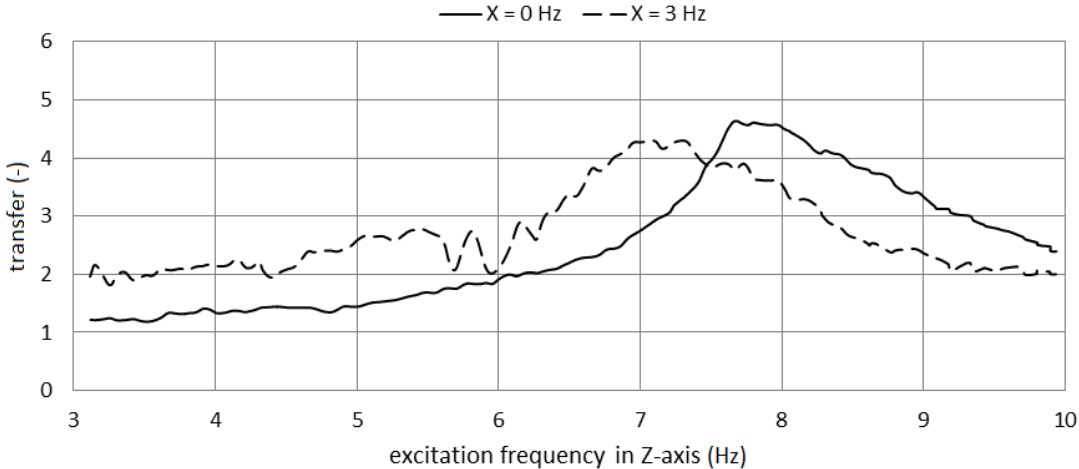

**Figure 11.** Transfer for Z-axis excitation 3 to 10 Hz and excitation in X-axis 3 Hz.

Figure 12 shows four transfer characteristics corresponding to simultaneous loading in vertical and horizontal directions. Horizontal loading was performed with excitation frequencies of 3, 6, and 9 Hz. The characteristic reduction of the resonance frequency and the resonance amplitude itself is clearly visible in all characteristics.

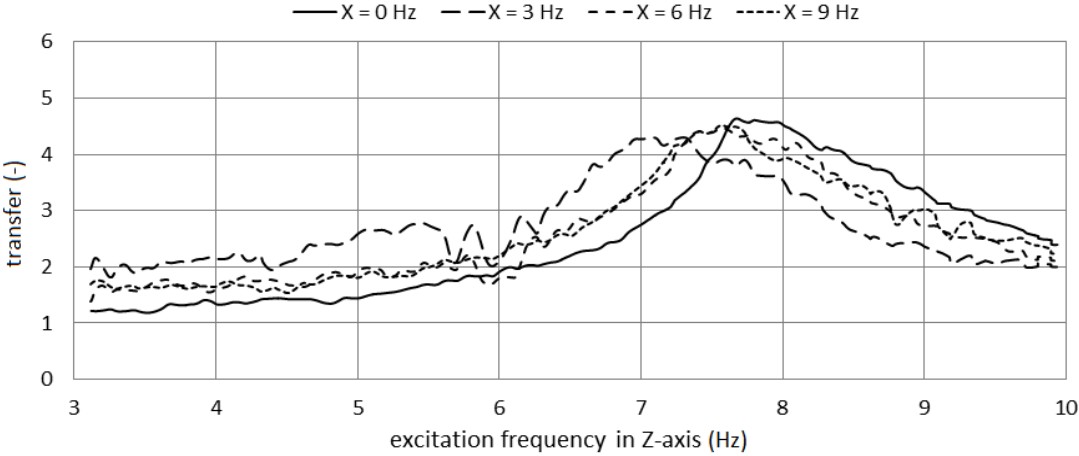

**Figure 12.** Transfer for Z-axis excitation 3 to 10 Hz and excitation in X-axis 3, 6, 9 Hz.

Table 1 summarizes the values of amplitudes and frequencies of resonance in the Z axis depending on the frequency of horizontal excitation. The results show that there are two effects if we introduce excitation in the X axis. The first effect is that the natural frequency is reduced. The second effect is that the transfer value is reduced. A more pronounced drop in transfer was seen at lower frequencies (a drop of 22.8% at 3 Hz and 8.5% at 6 Hz). At higher excitation frequencies, the drop was smaller, 2.1% for 9 Hz.

**Table 1.** Transfer drop and resonance frequency for Z-axis at excitation in Z-axis from 3 to 10 Hz and excitation in X-axis.

| Excitation X (Hz) | Transfer (-) | Transfer Drop (%) | Resonance Frequency in Z (Hz) |
|---|---|---|---|
| 0.0 | 4.7 | 0 | 7.8 |
| 3.0 | 4.1 | 22.8 | 7.1 |
| 6.0 | 4.3 | 8.5 | 7.5 |
| 9.0 | 4.6 | 2.1 | 7.6 |

The resonance in the X-axis occurs at a frequency of 3 Hz, this causes the excitation with a frequency of 3 Hz to have the greatest effect on the result of biaxial loading when there is a shift of the total resonant frequency from 7.8 to 7.1 Hz and also a decrease in the total transfer measured in the Z-axis. X-axis frequencies of 6 and 9 Hz are not in the resonance area, therefore, their influence on the shift of the total resonant frequency and the total transfer measured in the Z-axis is smaller and is located in the middle. The further we move away from the excitation frequency of 3 Hz, the influence of excitation in the X-axis on the total resonant frequency and total transfer will decrease.

The found resonance frequency of 3 Hz in the horizontal direction corresponds to the study [7], where the authors also arrived at a frequency of 3 Hz. In the case of the study [18], it was 6 Hz, which is due to the higher stiffness of the seat.

In the case of our seat, a resonance frequency of 7.8 Hz was measured in a purely vertical direction, studies [6,11] reached a value of 4 Hz, which is usually due to the effort to move in the comfort zone for the human body, but does not meet the requirements of seat manufacturers who require lighter seats with a thinner layer of PU foam, our higher resonance frequency value is due to the thinner layer of the seat.

The detected transfer value is 4.7. In the studies [15,18], they arrived at values of 4.0, which is due to the use of a weight in the form of a mannequin whose leaning on the backrest reduces the transfer value, a similar effect can be observed in the study [19], here, the transfer is 2.7 for human body load case. According to the study [20], the detected size of transfer with a separate weight is 4.8, which corresponds to our value of 4.7.

The results show the effect of horizontal excitation on the overall resonant frequency when it decreases. The reduction of the value due to the overall resonance corresponds to the findings in the study [19]. This reduction should not be neglected when designing a car seat.

The described effects of biaxial loading extend the knowledge gained from uniaxial loading described in the studies [16,17] using a device [4]. In addition to the above, the performed tests showed the functionality of the developed device [22], which makes it possible to carry out tests according to the relevant standards for biaxial loading of car seats [1–3]. The effect of biaxial loading on the transfer characteristic has been proven. Further research should include 3-axis testing. The designed equipment [5], which complements and exceeds current patented solutions [23–30], can be used for this purpose.

## 4. Conclusions

The aim of the research was to perform initial measurements on the developed device [22] and to compare the influence of one- and bi-axial dynamic loading of a car seat. The results showed a significant effect of horizontal excitation on the transfer in the car seat when both the transfer and the resonance of the car seat were reduced. In real operation, the car seat is exposed to multi-axial loads, for this reason, a 3-axis test device was designed [5], which should bring further refinement of the behavior of the car seat.

The subject of further work will be the execution of a more extensive set of measurements at different input parameters (amplitude, frequency) as well as a set of seats made of different foam stiffness and foam thickness. Knowledge of natural frequencies must be taken into account when designing the seats so that there is no unwanted strain on the human body of the crew.

**Author Contributions:** Conceptualization, V.F. and P.L.; methodology, A.L.; software, A.L.; validation, V.F., P.L. and A.L.; formal analysis, P.L.; investigation, V.F.; resources, P.L.; data curation, A.L.; writing—original draft preparation, V.F., P.L. and A.L.; writing—review and editing, P.L.; visualization, A.L.; supervision, V.F.; project administration, P.L.; funding acquisition, P.L. All authors have read and agreed to the published version of the manuscript.

**Funding:** This work was supported by the Ministry of Education, Youth and Sports of the Czech Republic and the European Union—European Structural and Investment Funds in the frames of Operational Programme Research, Development and Education—project Hybrid Materials for Hierarchical Structures (HyHi, Reg. No. CZ.02.1.01/0.0/0.0/16_019/0000843).

**Data Availability Statement:** The data presented in this study are available on request from the corresponding author. The data are not publicly available.

**Conflicts of Interest:** The authors declare no conflict of interest.

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
