# Peer review of "Effect of Dynamic Biaxial Loading of Car Seats"

_processes, doi:10.3390/pr10122483_

Round 1

Reviewer 1 Report

The article is at the required level in terms of formality, content and language and will be proposed for publication without further modifications. Congratulations on the eventual publication.

Author Response

Dear reviewer,

thank you for reading our paper and for your positive statement.

Best regards,

Authors 

Reviewer 2 Report

The author studies the influence of biaxial load on automobile seat comfort, which is beneficial to automobile seat design. The following suggestions are for the author's reference:

1)      Figure 10 shows that resonance occurs in the 7 Hz range. According to Figure 11 and Table 1, it should be more accurate near 7.8.

2)      When 3Hz, 6Hz and 9Hz are added to the X axis, the amplitude of 3Hz is the minimum, and the amplitude of 6 and 9Hz is in the middle, which can be explained briefly.

Author Response

Dear reviewer,

thank you for your comments, which help us to improve our paper. We have taken in account both your comments as you can see in the paper or bellow.

__ 

1)      Figure 10 shows that resonance occurs in the 7 Hz range. According to Figure 11 and Table 1, it should be more accurate near 7.8.

OUR EXPLANATION (IN PAPER IT IS ABOVE FIG. 10):

For comparison, tests were first performed with only vertical load in the Z axis in the frequency range from 1 to 16 Hz, which is the usual range for testing car seats Figure 10 shows that resonance occurs in the region between 7 - 8 Hz. This completely corresponds with the characteristic features of the measured seat. For further experiments, the measured band was narrowed to a range from 3 to 10 Hz, which sufficiently covers the interesting region around the resonance frequency. By narrowing the band, the experiment time and data volume were significantly reduced. Increasing the low frequency further made it possible to shorten the length of the window used for automatic amplitude detection. A narrower window allows for more accurate detection. Changing the detection parameters can cause even a small change in the calculated transfer function, so the results before and after the change cannot be compared. For all subsequent experiments, the settings remained the same so that the results were not affected and the effect of transverse excitation in the X-axis could be observed.

2)      When 3Hz, 6Hz and 9Hz are added to the X axis, the amplitude of 3Hz is the minimum, and the amplitude of 6 and 9Hz is in the middle, which can be explained briefly.

OUR EXPLANATION (IN PAPER UNDER THE TABLE 1):

The resonance in the X-axis occurs at a frequency of 3 Hz, this causes the excitation with a frequency of 3 Hz has the greatest effect on the result of biaxial loading, when there is a shift of the total resonant frequency from 7.8 to 7.1 Hz and also a decrease in the total transfer measured in the Z-axis. X-axis frequencies of 6 and 9 Hz are not in the resonance area, therefore their influence on the shift of the total resonant frequency and the total transfer measured in the Z-axis is smaller and is located in the middle. The further we move away from the excitation frequency of 3 Hz, the influence of excitation in the X-axis on the total resonant frequency and total transfer will decrease.

__  

We believe we met you demands.

Thank you.

Best regards,

Authors.